# Influence of Acute and Chronic Graft-Versus-Host Disease on Persistence of Antibodies against Measles, Mumps, Rubella and Varicella in the First Year after Autologous or Allogeneic Hematopoietic Stem Cell Transplantation

**DOI:** 10.3390/vaccines11030656

**Published:** 2023-03-14

**Authors:** Nicole Harrison, Heinz Burgmann, Werner Rabitsch, Claudia Honsig, Oliver Robak

**Affiliations:** 1Department of Medicine I, Division of Infectious Diseases and Tropical Medicine, Medical University of Vienna, 1090 Vienna, Austria; 2Department of Medicine I, Division of Bone Marrow Transplantation, Medical University of Vienna, 1090 Vienna, Austria; 3Department of Laboratory Medicine, Division of Clinical Virology, Medical University of Vienna, 1090 Vienna, Austria; 4Intensive Care Unit, Department of Medicine I, Medical University of Vienna, 1090 Vienna, Austria

**Keywords:** hematopoietic stem cell transplantation, live-attenuated vaccines, measles, mumps, rubella, varicella, immunity

## Abstract

Patients after hematopoietic stem cell transplantation (HSCT) are vulnerable to infections due to severe immunosuppression. Live-attenuated vaccines are contraindicated for two years after HSCT. The aim of this study was to assess the persistence of antibodies against measles, mumps, rubella and varicella in the first year after HSCT. Forty patients undergoing autologous (*n =* 12) or allogeneic (*n* = 28) HSCT were included in this study. Specific IgG antibodies to measles, mumps, rubella and varicella virus in serum samples were assessed by the LIAISON XL, a fully automated chemiluminescence analyzer, at seven different time points starting one week before HSCT and up to 12 months after HSCT. At baseline, before HSCT, most patients showed antibodies against measles (100%), mumps (80%), rubella (97.5%) and varicella (92.5%). Although titers declined over time, most patients retained antibodies against measles (92.5%), mumps (62.5%), rubella (87.5%) and varicella (85%) up to 12 months after HSCT. There was no significant difference between patients with and without GvHD concerning persistence of antibody titers. Significantly higher varicella titers were detected in autologous patients compared to patients with chronic GvHD. Considering that live-attenuated vaccines should not be administered during the first year after HSCT, the persistence of antibodies against these diseases is relevant.

## 1. Introduction

Hematopoietic stem cell transplantation (HSCT) is a process used to treat hematological malignancies, such as leukemia, lymphoma, myeloproliferative disorders and other diseases. Allogeneic HSCT means that stem cells are harvested from a healthy donor and then transferred to the patient, while for autologous transplantation previously harvested cells are returned to the same individual [1]. Once the donor immune system has engrafted, it takes a certain amount of time for the donor cells, especially the lymphocytes, to mature and differentiate before complete immune functions can be fulfilled. During this time, the patient is more vulnerable to various infections [2]. The process of immune reconstitution takes one to two years but can be considerably longer in patients with severe graft-versus-host disease (GvHD) [3,4,5]. The development of B-lymphocytes is very sensitive to the effects of GvHD and immunosuppressive therapy [6]. Therefore, patients with GvHD and prolonged immunosuppressive treatment are especially vulnerable to infections which require the formation of neutralizing antibodies as a first line of defense. Active immunization with vaccines is a valuable approach to prevent certain bacterial and viral infections after HSCT [7]. However, considering that lymphocytes need several months before they are mature enough to produce an effective vaccine response, the right timing for the vaccination of patients after HSCT is difficult to determine. Furthermore, it remains unclear to what extent the immunity of the donor against certain diseases is transferred to the patient [8,9]. Current guidelines recommend starting with inactivated vaccines as early as three months after transplantation [10,11]. However, live-attenuated vaccines are contraindicated for at least two years after transplantation or even longer if ongoing immunosuppressive therapy is necessary because of GvHD [12,13]. The most important live-attenuated vaccines are against the highly contagious diseases measles, mumps, rubella and varicella—all of which are part of the national immunization plan in Austria [14]. However, willingness to receive vaccination has declined during the last several decades, resulting in an increased incidence of diseases such as measles [15,16,17]. This decline in herd immunity against contagious diseases poses a threat to patients with severe immunosuppression [18]. There are studies which demonstrate that antibodies against measles, mumps and rubella persist after stem cell transplantation but then rapidly decline within the first few years [19,20]. It is unclear whether immunity is transferred from the donor. On the one hand, several studies provide evidence for the transfer of immunity from the donor to the recipient [21,22,23,24]. On the other hand, Ljungman et al. demonstrated that patients with natural measles infection retained positive titers against measles for much longer than patients who were vaccinated before stem cell transplantation [20]. The persistence of antibodies against diseases, such as measles, mumps, rubella and varicella, is essential during the first year after stem cell transplantation, as vaccination is contraindicated, and patients are vulnerable to these infections.

The aim of this study was to gain further knowledge about the persistence of antibodies against the above-mentioned diseases during the first year after autologous and allogeneic HSCT and about the influence of acute and chronic graft-versus-host disease on persisting immunity.

## 2. Materials and Methods

### 2.1. Study Design and Study Population

This study was approved by the Ethics Committee of the Medical University of Vienna (registration number: 1578/2017). The analysis was performed on historical serum samples. Between November 2008 and December 2010, patients undergoing autologous or allogeneic hematopoietic stem cell transplantation at the Department for Bone Marrow Transplantation at the University Hospital of Vienna were prospectively included. We enrolled patients prior to the start of myeloablative or reduced-intensity conditioning (RIC) for allogeneic HSCT (*n =* 28) and autologous HSCT patients (*n =* 12) as controls. All patients gave written informed consent prior to enrollment. Patients with the myeloablative conditioning regimen (*n =* 26) received cyclophosphamide and full body irradiation with 13.2 Gy. Patients with RIC (*n =* 14) were assigned according to their disease status, age and comorbidities. Patients with complete remission (CR) of acute myeloid leukemia received the Seattle protocol (fludarabine, irradiation with 2 Gy, *n =* 6), whereas patients without CR were assigned to the FLAMSA protocol (fludarabine, amsacrin, cytarabin, *n =* 4). Patients with chronic lymphatic leukemia were given fludarabine and cyclophosphamide (*n =* 1), patients with Hodgkin’s lymphoma received BEAM (carmustine, etoposide, cytarabine, melphalan, *n =* 2) and patients with myelodysplastic syndrome received busulfan, fludarabine and ATG (*n =* 1). Serum samples were collected at the following time points: on admission (7 days before HSCT; T − 1), on the day of HSCT (T0), during aplasia (absolute neutrophil count <0.5 G/L; T + 1), at engraftment (T + 2; defined by a neutrophil count >0.5 G/L), 1 month after HSCT (T + 3), 3 to 6 months after HSCT (T + 4), and 6 to 12 months after HSCT (T + 5). The severity of chronic GvHD was defined based on the modified Glucksberg and NIH classification [4]. Grading of acute GvHD was based on the severity of skin involvement (extension of maculopapular rash and/or bullous lesions), serum bilirubin levels and severity of diarrhea, as defined by Glucksberg [25]. All values were measured at the time point closest to maximum GvHD activity. Time-matched samples were used for the comparison of GvHD and non-GvHD patients. All patients received anti-infective prophylaxis, as described in [7]. None of the patients received immunoglobulins during the study period. The immunosuppressive therapy for acute GvHD (aGvHD) included steroids (2 mg/kg/day), calcineurin inhibitors and, for salvage therapy, extracorporeal photopheresis (ECP). Similarly, for treatment of chronic GvHD (cGvHD), steroids in combination with calcineurin inhibitors and ECP were administered.

### 2.2. Serology

Specific IgG antibodies to measles, mumps, rubella and varicella zoster virus in serum and plasma samples were assessed on the LIAISON ^®^ XL, a fully automated chemiluminescence analyzer (DiaSorin, Saluggia, Italy). Both the LIAISON^®^ measles and mumps IgG assays use indirect sandwich chemoluminescence immunoassay (CLIA) technology for the semi-quantitative determination of specific IgG antibodies. Concentrations of the specific antibodies are given as arbitrary units/mL. The LIAISON^®^ rubella and varicella zoster Virus IgG assays use indirect chemoluminescence immunoassay (CLIA) technology for the quantitative assessment of the respective antibody concentrations. The concentrations of specific rubella IgG are reported as IU/mL, those of specific varicella zoster virus IgG as mIU/mL. All tests were performed according to the manufacturer’s instructions. The results were classified as negative, borderline or positive, according to the cutoff values established by the department of clinical virology (positive values: measles ≥ 20 AU/mL, mumps ≥ 20 AU/mL, rubella ≥ 15 IE/mL, varicella ≥ 165 mIU/mL). Chemoluminescence immunoassays assess the presence of antibodies against the above-mentioned antigens but not their neutralizing capacity. In the clinical routine, neutralizing assays are not utilized, and therefore positive antibody results measured by ELISA or CLIA are usually interpreted as immunity against the disease [26].

### 2.3. Statistical Analysis

The study was designed as a retrospective analysis of serum samples from a prospective cohort including 40 patients. Non-relapse mortality (NRM) was defined as any death not related to the underlying malignancy. Relapse was defined as recurrence of malignancy after achievement of complete remission. Cumulative incidences of aGvHD and cGvHD were estimated, considering relapse and death as competing events. Overall survival was calculated from day 0 of HSCT to the day of death from any cause or the last follow-up. Statistical comparisons of titers in between-patient cohorts were made using the unpaired Student’s *t*-test or the Mann–Whitney *U* test following testing for normality. Fisher’s exact test was used to examine the significance of the association between two variables. Predictors for positive antibody titers were calculated using univariate logistic regression. Differences were considered statistically significant at a two-sided *p*-value of <0.05. The data are presented as means and standard deviations. All data were calculated using GraphPad Prism 6.0 (GraphPad Software, San Diego, CA, USA).

## 3. Results

### 3.1. Demographics of the Study Population

Overall, 40 patients undergoing autologous (*n =* 12, 10 male) or allogeneic (*n =* 28, 11 male) hematopoietic stem cell transplantation were included in this study. Apart from gender and disease, no significant differences in patient characteristics between the autologous and the allogeneic HSCT patient cohorts were observed. The characteristics of the study population are included in Table 1. Cumulative incidence of acute GvHD was 65% (*n =* 18) and that of chronic GvHD was 61% (*n =* 17), as depicted in Table 2.

### 3.2. Time Course of Antibody Titers against Measles, Mumps, Rubella and Varicella

At baseline, before HSCT, most patients showed antibodies against measles (*n =* 40, 100%), mumps (*n =* 32, 80%), rubella (*n =* 39, 97.5%) and varicella (*n =* 37, 92.5%). The majority of patients retained antibodies against measles (*n =* 37/40, 92.5%, allogeneic: *n =* 25/28, autologous: *n =* 12/12), mumps (*n =* 25/40, 62.5%, allogeneic: *n =* 15/28, autologous: *n =* 10/12), rubella (*n =* 35/40, 87.5%, allogeneic: *n =* 24/28, autologous: *n =* 11/12) and varicella (*n =* 34/40, 85%, allogeneic: *n =* 23/28, autologous: *n =* 11/12) up to 12 months after HSCT. The time courses of mean antibody titers against measles, mumps, rubella and varicella from T – 1 to T + 5 are shown in Figure 1. Measles titers in allogeneic HSCT recipients declined from 254.1 (±92.2.0) AU/mL at T – 1 to 201.5 (±107.9) AU/mL at T5, but this decline did not reach statistical significance (*p* = 0.476). Mumps titers (83.5 (±80.3) AU/mL at T – 1 to 41.7 (±53.1) AU/mL at T5; *p* = 0.219), rubella titers (62.7 (±74.0) IU/mL at T – 1 to 45.8 (±48.8) IU/mL at T5; *p* = 0.378) and varicella titers (1191.1 (±930.3) mIU/mL at T – 1 to 1074.5 (±902.3) mIU/mL at T5; *p* = 0.789) also did not decline significantly over 12 months after HSCT. Antibody titers did not differ significantly between patients after allogeneic and autologous HSCT. At all seven time points, absolute lymphocyte counts were assessed and did not differ significantly between the allogeneic and autologous groups (Appendix A).

### 3.3. Influence of Acute GvHD on Antibody Titers in Allogeneic Patients

Next, the influence of acute GvHD on antibody titers was assessed by comparing allogeneic patients with and without GvHD (titers at the time point of the highest severity of acute GvHD compared to the same time point in patients without GvHD). In allogeneic HSCT recipients with aGvHD, mean measles titers were 248.2 ± 99.7 AU/mL and were not significantly different compared to allogeneic HSCT patients without aGvHD (266.9 ± 77.1 AU/mL; *p* = 0.447; Figure 2). Antibody titers against mumps (67.7 ± 55.1 AU/mL with aGvHD vs. 35.3 ± 39.5 AU/mL without aGvHD; *p* = 0.091), rubella (37.6 ± 38.8 IU/L with aGvHD vs. 46.1 ± 22.8 IU/L without aGvHD; *p* = 0.453) and varicella (1057.0 ± 959.8 mIU/mL with aGvHD vs. 867.4 ± 605.2 mIU/mL without aGvHD; *p* = 0.398) were not significantly different when comparing patients with and without aGvHD. Antibody titers of patients with aGvHD were not significantly different compared to patients after autologous HSCT. In addition, the antibody titers at baseline (T − 1) were compared to the time point with the highest severity of acute GvHD (T2 or 3) in patients with GvHD. There was no significant difference between the antibody titers at baseline or at the highest severity of acute GvHD (Appendix A).

### 3.4. Influence of cGvHD on Antibody Titers in Allogeneic Patients

The influence of cGvHD on antibody titers was assessed by comparing allogeneic patients with and without chronic GvHD (titers at the time point of highest severity of chronic GvHD compared to the same time point in patients without GvHD). In allogeneic HSCT recipients with cGvHD, mean measles titers were 213.0 (±109.1) AU/mL and not significantly different compared to HSCT patients without cGvHD (247.1 ± 96.44 AU/mL; *p* = 0.594). Antibody titers against mumps (54.9 ± 58.5 AU/mL with cGvHD vs. 38.2 ± 41.2 AU/mL without cGvHD; *p* = 0.599), rubella (50.0 ± 52.9 IU/L with cGvHD vs. 40.2 ± 45.3 IU/L without cGvHD; *p* = 0.590) and varicella (779.7 ± 576.8 mIU/mL with cGvHD vs. 1159.0 ± 1151.2 mIU/mL without cGvHD; *p* = 0.564) were not significantly different when comparing patients with and without cGvHD. Antibody titers against varicella of patients with cGvHD were significantly lower compared to patients after autologous HSCT (1645.1 ± 747.9 mIU/mL; *p* = 0.002; Figure 3). In addition, the antibody titers at baseline (T − 1) were compared to the time point of the highest severity of chronic GvHD (T4 or 5) in patients with GvHD. There was no significant difference between the antibody titers at baseline and at the highest severity of chronic GvHD (Appendix A).

### 3.5. Influence of Severity of aGvHD and cGvHD on Antibody Titers

Severity of aGvHD correlated with antibody titers only in the case of mumps titers (aGvHD grade 3 vs. no GvHD *p* = 0.038; Figure 4). Severity of cGvHD did not correlate significantly with respective antibody titers.

### 3.6. Influence of Age, Gender and Conditioning Regimen on Antibody Titers

In a univariate linear regression model with the respective titers at T + 5 as dependent variables, increasing age and female gender correlated with higher measles titers, but not conditioning regimen, donor type or donor mismatch (Table 3). None of these variables correlated with mumps, rubella or varicella titers at T + 5.

### 3.7. Survival

Overall, 13 patients (32%) died within a mean of 8 months after HSCT. Nine patients (22.5%) died within a mean of 4.7 months after HSCT due to relapse of their primary diseases. Four patients (10%) died within a mean of 14.5 months after HSCT due to causes that were not related to their primary disease (causes of death: sarcoma, infection, cGvHD and aGvHD). We did not observe differences in mortality, relapse, aGvHD or cGvHD with regard to gender.

## 4. Discussion

This study focused on the persistence of antibodies during the process of and the first year after HSCT when complications such as acute GvHD can drastically influence the functions of the immune system [1]. Several studies have described the decline in antibodies in the years after HSCT [19,20], but there are no data for the first year. This study aspired to find fluctuations or changes in antibody titers during conditioning, aplasia, engraftment, and during periods of acute or chronic GvHD. Interestingly, although antibody titers declined over the 12 months after HSCT, these changes were not statistically significant. Moreover, there were no sudden changes during aplasia or during periods of acute GvHD. The majority of patients retained antibodies against measles, mumps, rubella and varicella during the first year after HSCT. This is of importance, as re-vaccination with live-attenuated vaccines is not possible during this period of severe immunosuppression.

The most important of these diseases regarding morbidity and mortality is measles, of which there have been recent outbreaks in several countries [27,28,29]. The immunity for measles was very high in this study population, with 100% seropositivity before HSCT, and only three patients became seronegative during the first year. This is a positive result, considering that many patients are protected against measles. Patients after HSCT or with hematological malignancies who are exposed to measles might receive immunoglobulin treatment to prevent infection. However, considering these results, measles titers should be determined beforehand, if feasible, as many patients retain immunity [30].

Early vaccination after HSCT is another possibility to protect patients. Several studies have provided evidence that vaccination against measles is safe and effective in the second year after HSCT [31]. One study by Machado et al. [32] vaccinated as early as 9 months after HSCT during a measles outbreak and showed high seroconversion rates in non-immune patients but overall low increases in antibody titers. There is still limited evidence concerning patients with ongoing immunosuppressive therapy and chronic GvHD. Our study showed that, during the first year after HSCT, post-transplant titers persisted in most patients. Therefore, vaccination in the second year after HSCT might be a good strategy to provide early but safe vaccination in patients without GvHD.

The influence of acute and chronic GvHD was assessed by matching allogeneic patients with and without GvHD and comparing their respective antibody titers. There was no influence of either acute or chronic GvHD observed in allogeneic patients. The development of GvHD is usually accompanied by an increase in immunosuppressive therapy. Although therapeutic regimens were not assessed independently, it can be said that the immunosuppression due to acute GvHD did not impact the persistence of antibodies against these diseases. Other studies have reported that acute GvHD is a risk factor for seronegativity for measles after HSCT [19]. Autologous patients showed significantly higher titers than patients with chronic GvHD for varicella but not for any of the other diseases. There was no significant difference between allogeneic patients with and without chronic GvHD concerning varicella antibodies, although the titers were higher in the latter group. Therefore, chronic GvHD showed a certain influence on the persistence of varicella antibodies, especially in comparison with autologous patients. Varicella is a virus that can be reactivated and cause herpes zoster disease in patients. In this patient group, no reactivation of varicella zoster virus was documented in the allogeneic or autologous groups. Therefore, this did not influence the persistence of varicella titers after HSCT. So far, we have no explanation for this observation, but it might indicate that chronic GvHD influences the persistence of antibodies against varicella. It is unclear whether patients with chronic GvHD are more vulnerable to varicella infection.

Considering our small sample size, it is possible that this study underestimated the effects of GvHD. Another possible explanation is that patients with GvHD retain immunity during the first year but are more prone to becoming seronegative in the following years after HSCT.

The influence of other factors, such as age, gender, conditioning regimen, donor type and donor mismatch, on the persistence of antibody titers was assessed. Female patients had significantly higher antibody titers against measles (*p* = 0.042) than male patients. Female gender is often associated with a stronger antibody response to vaccination [33]. Considering that the autologous control group was predominantly male (10 out of 12 patients), this might have influenced our results to a certain degree. Increasing age correlated with increasing antibody titers for measles (*p* = 0.036). Immunity to vaccine-preventable diseases usually declines with age, necessitating shorter booster intervals. However, it was reported that immunity against measles persisted longer in patients who experienced natural measles infection (often in the older age group) compared to patients who were vaccinated [19]. In this study, we were not able to obtain information on vaccination status or previous infections due to the retrospective design.

Possible strengths of the study are the focus on the first year after HSCT, with several time points being assessed, allowing for a closer analysis of the persistence of antibodies in this period, and the autologous patients as a control group to account for the different conditioning regimens and the lack of GvHD in this group of patients. Possible weaknesses of the study are the small number of patients in each group, the predominantly male control group, the retrospective design and the lack of information on vaccination status.

## 5. Conclusions

In conclusion, this study shows that antibodies against measles, mumps, rubella and varicella persist in most patients in the first year after HSCT. There was no significant difference between patients with acute or chronic GvHD and patients without GvHD concerning persistence of antibody titers. Notably, significantly higher varicella titers were detected in autologous patients compared to patients with chronic GvHD. Patients cannot be vaccinated with a live-attenuated vaccine in the first year after HSCT, but this study showed, in a relatively small sample, that immunity is often retained within this time frame.

## Figures and Tables

**Figure 1 vaccines-11-00656-f001:**
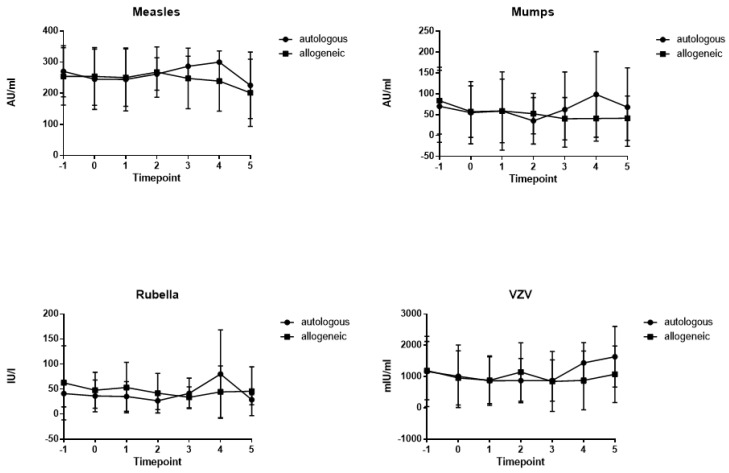
Time course of mean antibody titers for measles, mumps, rubella and varicella from time points T − 1 to T + 5 for allogeneic (squares) and autologous (circles) patients (lines depict standard deviations). Time points: on admission (7 days before HSCT; T − 1), on the day of HSCT (T0), during aplasia (absolute neutrophil count < 0.5 G/L; T + 1), at engraftment (T + 2, neutrophil count > 0.5 G/L), 1 month after HSCT (T + 3), 3 to 6 months after HSCT (T + 4), and 6 to 12 months after HSCT (T + 5).

**Figure 2 vaccines-11-00656-f002:**
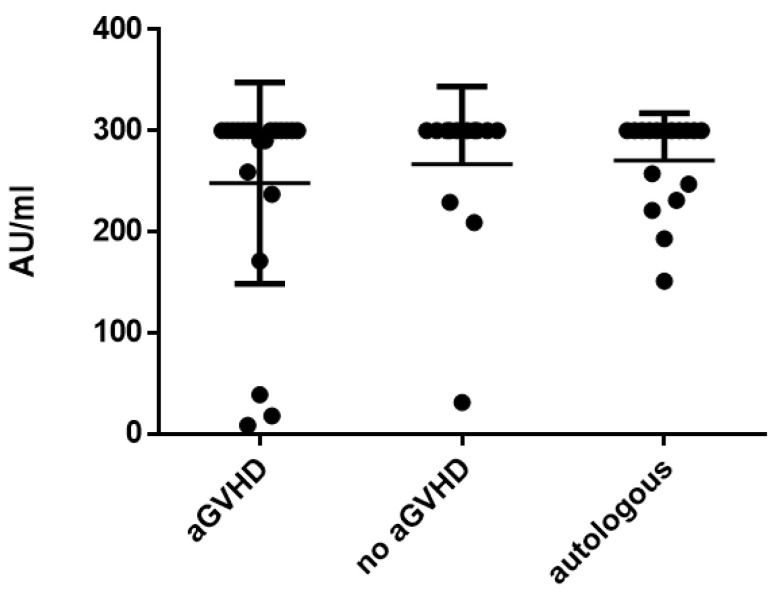
Mean measles titers comparing patients with acute GvHD vs. non-GvHD vs. autologous patients.

**Figure 3 vaccines-11-00656-f003:**
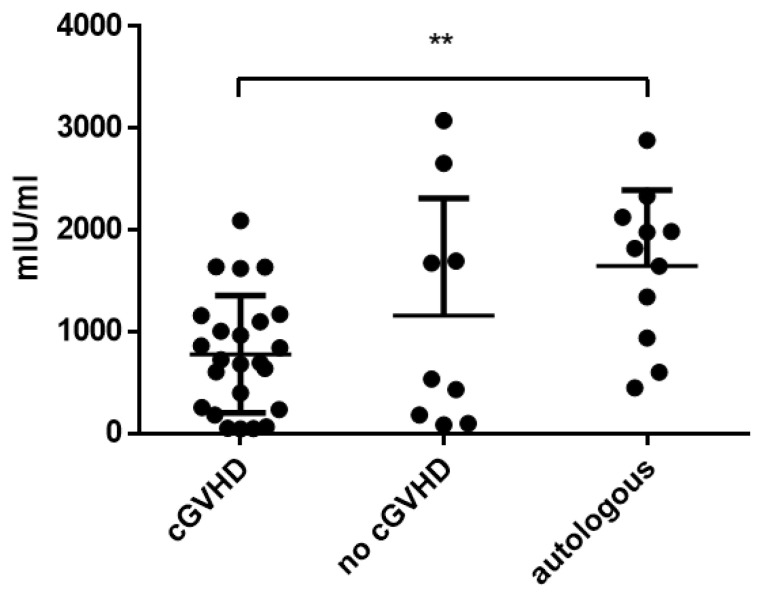
Mean varicella titers comparing patients with chronic GvHD vs. non-GvHD vs. autologous patients. Asterisks mark significant *p*-values.

**Figure 4 vaccines-11-00656-f004:**
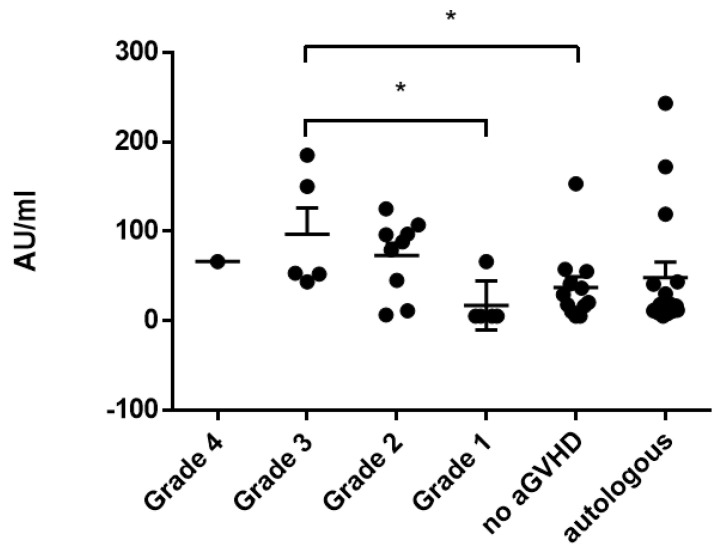
Correlation of mumps titers and severity of acute GvHD (grades 1–4) compared to patients without GvHD and autologous patients. Asterisks mark significant *p*-values.

**Table 1 vaccines-11-00656-t001:** Demographic data of allogeneic and autologous patients.

	All	Allogeneic	Autologous
	N (%)	N (%)	N (%)
**Number of patients**	40 (100)	28 (70)	12 (30)
**Median age in years (range)**	46 (34–56)	44 (34–56)	48 (42–49)
**Gender**			
Male	21 (53)	11 (39)	10 (83)
Female	19 (47)	17 (61)	2 (17)
**Mean BMI**	24.4	22.9	25.8
**Diagnosis**			
Acute leukaemia	21 (53)	21 (75)	0 (0)
Chronic leukaemia	1 (3)	1 (4)	0 (0)
Lymphoma	9 (23)	5 (18)	4 (33)
Myeloma	6 (15)	0 (0)	6 (50)
Other †	3 (8)	1 (4)	2 (17)
**Disease status at transplantation**			
Standard risk §	21 (53)	15 (54)	6 (50)
High risk §	19 (48)	13 (46)	6 (50)
**Conditioning regimen**			
Myeloablative	26 (65)	14 (50)	12 (100)
RIC	14 (35)	14 (50)	0 (0)
**Stem cell donors**			
Related	11 (39)	11 (39)	NA
Unrelated	17 (61)	17 (61)	NA
HLA-identical	21 (75)	21 (75)	NA
HLA-mismatched	7 (25)	7 (25)	NA
**Stem cell source**			
Bone marrow	1 (3)	1 (4)	0 (0)
PBSC	39 (98)	27 (96)	12 (100)
**Post-transplant immunosuppressive prophylaxis**			
Cyclosporine only	4 (10)	4 (14)	NA
Cyclosporine-MTX	14 (35)	14 (50)	NA
Cyclosporine-MMF	10 (25)	10 (36)	NA
**Median follow-up in months (range)**	26 (0.1–46)	24 (0.1–46)	30.3 (4.3–46)

BMI—body mass index; HLA—human leukocyte antigen; RIC—reduced intensity conditioning; PBSC—peripheral blood stem cells; MTX—methotrexate; MMF—mycophenolate-mofetil. † Other diagnoses included myelodysplastic syndrome and chronic lymphocytic leukemia. § Standard risk was defined as acute leukemia in the first or second complete remission or chronic myeloid leukemia in the first chronic phase. High-risk disease included myelodysplastic syndrome, acute and chronic leukemia beyond second complete remission or in relapse, as well as chronic myeloid leukemia beyond the first chronic phase.

**Table 2 vaccines-11-00656-t002:** Characteristics of acute and chronic graft-versus-host disease (GvHD) in allogeneic patients (*n =* 28).

	Acute GvHD N (%)	Chronic GvHD N (%)
Total	18 (64)	17 (61)
Organ involvement		
Skin	14 (78)	11 (65)
Eyes		11 (65)
Oral mucosa		8 (47)
Liver	10 (56)	10 (56)
Lungs		4 (24)
Gastrointestinal	10 (56)	2 (12)
Joints		1 (6)
Severity score (maximum)		
1	6 (33)	10 (59)
2	7 (39)	4 (24)
3	4 (22)	3 (18)
4	1 (6)	
Onset type of cGvHD		
De novo		6 (35)
Quiescent		6 (35)
Progressive		5 (29)
Median time to first onset of GvHD in days (range)	27 (10–80)	123 (75–222)

**Table 3 vaccines-11-00656-t003:** Influence of age, gender, conditioning regimen, donor type and donor mismatch on antibody titers at T + 5 by univariate logistic regression. Asterisks mark significant differences.

	Measles	Mumps	Rubella	Varicella
Age	0.036 *	0.445	0.532	0.530
Gender	0.042 *	0.343	0.411	0.298
Conditioning regimen	0.421	0.844	0.540	0.109
Donor type	0.629	0.897	0.129	0.649
Donor mismatch	0.157	0.329	0.197	0.124

## Data Availability

Data are available upon request from the corresponding author.

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
