# Peer review of "Influence of Acute and Chronic Graft-Versus-Host Disease on Persistence of Antibodies against Measles, Mumps, Rubella and Varicella in the First Year after Autologous or Allogeneic Hematopoietic Stem Cell Transplantation"

_vaccines, 2023, doi:10.3390/vaccines11030656_

Round 1

Reviewer 1 Report

This manuscript, written by Dr. Harrison, original research, with the title of "Influence of acute and chronic Graft-versus-Host-Disease on persistence of antibodies against Measles, Mumps, Rubella and Varicella in the first year after autologous or allogeneic hematopoietic stem cell transplantation" analyzed the antibody titers of measles, mumps, rubella, ad varicella in patients with hematopoietic stem cell transplantation (HSCT).

The research found that the serological values did not change significantly over time after the transplantation (Figure 1). Additionally, as shown in Figure 2, no differences were found between acute GVHD and non-GVHD.

Because patients with HSCT cannot be vaccinated with live-attenuated vaccines during the first year, it is important to know how the the antibody levels change over time. Overall, the data indicates the levels are kept.

The manuscript is well written, it is easy to read, and to understand.

The limitation of the sample size is acknoledged by the authors.

Comments:

1) Were tha patients vaccinated for those virus previously to the HSCT?

2) In Figure 1, the time course is shown. The levels of antibody at the beginning (time -1) are important. Are those levels "protective" against virus infection?

3) In Figure 2, the antibody titers were compared between the different groups at a particular point. Did you compare the levels of the same patient before and after developing aGVHD? For example, you could use a non parametric test, related samples.

4) Why levels in cGVHD were lower than autologous? Does it provide any clinical relevance?

5) Should patients receive vaccination previously to HSCT (some months previously)?

6) Could you please comment on this manuscript:

Groeneweg L, Loeffen YGT, Versluys AB, Wolfs TFW. Safety and efficacy of early vaccination with live attenuated measles vaccine for hematopoietic stem cell transplant recipients and solid organ transplant recipients. Vaccine. 2021 Jun 8;39(25):3338-3345. doi: 10.1016/j.vaccine.2021.04.049. Epub 2021 May 12. PMID: 33992440.   7) What about the safey of mRNA vaccines?

Author Response

1) Were the patients vaccinated for those viruses previously to the HSCT?

Due to the retrospective design of the study, we were not able to obtain information concerning the vaccination status of the patients. This is mentioned as a limitation of the study in the Discussion section (line 319-321).

2) In Figure 1, the time course is shown. The levels of antibody at the beginning (time -1) are important. Are those levels "protective" against virus infection?

The antibody levels at baseline (T -1) and the percentage of patients with positive antibody results are described in the results section (line 163/164). The absolute level of IgG antibodies cannot be correlated with the strength of immunity, there is no "correlate of protection". Specific IgG antibodies are a surrogate marker for humoral immunity, for cellular immunity we have no routine measurement methods available. Therefore, according to the manufacturer’s recommendations, we interpret positive IgG antibodies of any concentration above the validated cut-off (line 112/113) as indicative of immunity and indicative of protection against virus infection, respectively. Positive antibodies indicate earlier contact (immunization or infection) and will be boosted upon contact with the virus, regardless of the concentration measured.

Change in manuscript (line 122 to 126):

Chemoluminescence immunoassays assess the presence of antibodies against before-mentioned antigens, but not their neutralizing capacity. In the clinical routine neutralizing assays are not utilized and therefore positive antibody results measured by ELISA or CLIA are usually interpreted as immunity against the disease.

3) In Figure 2, the antibody titres were compared between the different groups at a particular point. Did you compare the levels of the same patient before and after developing aGvHD? For example, you could use a non parametric test, related samples.

We thank the reviewer for the suggestion and analysed the antibody titers at baseline and at time of highest severity of acute or chronic GvHD and did not find any significant changes.

Change in manuscript (line 199-202, 219-222, Supp. Figure 2 and 3):

In addition, the antibody titers at baseline (T -1) were compared to the time point with highest severity of acute GvHD (T2 or 3) in patients with GvHD. There was no significant difference between the antibody titers at baseline and at highest severity of acute GvHD (Supp. Figure S2).

In addition, the antibody titers at baseline (T -1) were compared to the time point with highest severity of chronic GvHD (T4 or 5) in patients with GvHD. There was no significant difference between the antibody titers at baseline and at highest severity of chronic GvHD (Supp. Figure S3).

4) Why levels in cGVHD were lower than autologous? Does it provide any clinical relevance?

The antibody levels for varicella were lower in patients with chronic GvHD than in autologous patients. So far, we cannot present an explanation for this observation, because in both patient groups no one experienced episodes of Herpes zoster illness (reactivation of varicella). The clinical relevance could be that patients with chronic GvHD are more vulnerable to varicella infection as antibody titres are lower after HSCT. However, our data is not sufficient to support this conclusion.

Change in manuscript (line 295-98):

So far, we have no explanation for this observation, but it might indicate that chronic GvHD influences the persistence of antibodies against varicella. It is unclear if patients with chronic GvHD are more vulnerable to varicella infection.

5) Should patients receive vaccination previously to HSCT (some months previously)?

Routine vaccination against MMRV before HSCT is not recommended and would be problematic as live-attenuated vaccines should not be administered to immunosuppressed patients. Most patients before HSCT are already immunosuppressed due to chemotherapy and the underlying disease. There is some discussion concerning donor vaccination as this could improve the vaccine response in the recipient. However, due to ethical concerns this is not feasible in most situations. Live-attenuated vaccines should not be given four weeks before donation because of the risk of vaccine-disease transmission.

Cordonnier, C.; Einarsdottir, S.; Cesaro, S.; Di Blasi, R.; Mikulska, M.; Rieger, C.; de Lavallade, H.; Gallo, G.; Lehrnbecher, T.; Engelhard, D.; et al. Vaccination of haemopoietic stem cell transplant recipients: guidelines of the 2017 European Conference on Infections in Leukaemia (ECIL 7). The Lancet. Infectious diseases 2019, 19, e200-e212, doi:10.1016/s1473-3099(18)30600-5.

6) Could you please comment on this manuscript:

Groeneweg L, Loeffen YGT, Versluys AB, Wolfs TFW. Safety and efficacy of early vaccination with live attenuated measles vaccine for hematopoietic stem cell transplant recipients and solid organ transplant recipients. Vaccine. 2021 Jun 8;39(25):3338-3345. doi: 10.1016/j.vaccine.2021.04.049. Epub 2021 May 12. PMID: 33992440.

We thank the reviewer for this valuable comment as early vaccination after HSCT is an important aspect. Most studies mentioned in this review vaccinated patients in the second year after HSCT, which was safe and efficient (seroconversion rates of up to 100%). One study by Machado et al. vaccinated as early as 9 months after HSCT. However, this was the only study which included patients with immunosuppression and chronic GvHD. Therefore, there is still limited evidence if live-attenuated vaccines are safe in patients with chronic GvHD. Our study showed that during the first year after HSCT post-transplant titres persisted in most patients. Therefore, vaccination in the second year after HSCT might be a good strategy to provide early but safe vaccination in patients without GvHD. We added this to the Discussion section of our manuscript.

Change in manuscript (line 271 to 279):

Early vaccination after HSCT is another possibility to protect patients after HSCT. Several Studies provide evidence that vaccination against measles is safe and effective in the second year after HSCT. One study by Machado et al. vaccinated as early as 9 months after HSCT during a measles outbreak and showed high seroconversion rates in non-immune patients, but overall low increases in antibody titres. There is still limited evidence concerning patients with ongoing immunosuppressive therapy and chronic GvHD. Our study showed that during the first year after HSCT post-transplant titres persisted in most patients. Therefore, vaccination in the second year after HSCT might be a good strategy to provide early but safe vaccination in patients without GvHD.

7) What about the safey of mRNA vaccines?

To our knowledge, there are currently no mRNA vaccines against measles, mumps, rubella or varicella. Therefore, we did not comment on this technology in our paper. Of course, new vaccination strategies other than live-attenuated vaccines would be favourable for this group of patients.

Reviewer 2 Report

Harrison et al. paper describes titration of antibody titers against measles, mumps, rubella and varicella in a series of patients undergoing autologous and allogeneic transplantation. 

The authors report 12 autologous transplant patients and 28 allogeneic transplant patients. 

The abstract section is concise and summarizes the results of the study, the introduction section is focused on the objectives of the study, the results section is divided into clearly defined subchapters. The discussion section comments on the results of the study.

Overall the impression of the study is positive.

Problems:

1) the authors should define the drugs and doses administered to patients in the conditioning regimen,

2) it is not clear whether patients had received replacement immunoglobulins during the study, in case of administration the authors should report the timing in the methods and results,

3) The authors should also report the values of peripheral blood lymphocytes when determining antibody titres,

4) the authors should also clarify that the results obtained relate to the presence of antibodies in the peripheral blood of patients and not necessarily to their neutralizing capacity,

5) The authors should check the dots in the figures along the text.

Author Response

1) the authors should define the drugs and doses administered to patients in the conditioning regimen,

We added the different conditioning regimens to our manuscript.

Change in manuscript (line 84-93):

Patients with myeloablative conditioning regimen (n=26) received cyclophosphamide and full body irradiation with 13.2 Gy. Patients with RIC (n=14) were assigned according to their disease status, age and comorbidities. Patients with complete remission (CR) of acute myeloid leukemia received the Seattle protocol (fludarabine, irradiation with 2 Gy, n=6), whereas patients without CR were assigned to the FLAMSA protocol (fludarabine, amsacrin, cytarabin, n=4). Patients with chronic lymphatic leukemia were given fludarabine and cyclophosphamide (n=1), patients with Hodgkin’s lymphoma received BEAM (carmustine, etoposide, cytarabine, melphalan, n=2) and with myelodysplastic syndrome received busulfan, fludarabine and ATG (n=1).

2) it is not clear whether patients had received replacement immunoglobulins during the study, in case of administration the authors should report the timing in the methods and results,

Patients did not receive immunoglobulins during the study period.

Change in manuscript (line 103):

None of the patients received immunoglobulins during the study period.

3) The authors should also report the values of peripheral blood lymphocytes when determining antibody titres,

We assessed the absolute lymphocyte counts at all seven time points and displayed the values in Supplemental figure 1.

Change in manuscript (line 177-79 and Supplemental Figure 1):

At all seven time points absolute lymphocyte counts were assessed and did not differ significantly between the allogeneic and autologous group (Supp. Figure S1).

4) the authors should also clarify that the results obtained relate to the presence of antibodies in the peripheral blood of patients and not necessarily to their neutralizing capacity,

We agree with the reviewer that our results report the presence of antibodies but not necessarily their neutralizing capacity. A neutralization assay would be necessary to confirm that the present antibodies are neutralizing. However, in the clinical routine neutralizing assays are not utilized and therefore positive antibody results measured by ELISA or CLIA (chemoluminescence immunoassay) are usually interpreted as immunity against the disease. It is known that IgG antibody concentrations detected in ELISA or CLIA correlate with the concentrations measured in neutralization assays (Rabenau et al, 2007). It is also generally accepted that ligand assays are suitable for immunity testing. We added an explanation to the methods section.

Rabenau, H.F., Marianov, B., Wicker, S. et al. Comparison of the neutralizing and ELISA antibody titres to measles virus in human sera and in gamma globulin preparations. Med Microbiol Immunol 196, 151–155 (2007). https://doi.org/10.1007/s00430-007-0037-2

Change in manuscript (line 122 to 126):

Chemoluminescence immunoassays assess the presence of antibodies against before-mentioned antigens, but not their neutralizing capacity. In the clinical routine neutralizing assays are not utilized and therefore positive antibody results measured by ELISA or CLIA are usually interpreted as immunity against the disease.

5) The authors should check the dots in the figures along the text.

We checked all dots in the figures and compared them to the reported results in the text and did not find any inconsistencies.

Reviewer 3 Report

Nicole Harrison and colleagues present a high quality and well-written experimental manuscript reporting influence of acute and chronic graft-versus-host-disease on persistence of antibodies against measles, mumps, rubella and varicella in the first year after autologous or allogeneic hematopoietic stem cell transplantation.

Authors aimed to assess the persistence of antibodies against measles, mumps, rubella and varicella in the first year after HSCT. For that 40 patients undergoing autologous (n=12) or allogeneic (n=28) HSCT were included in this study. Specific IgG antibodies to measles, mumps, rubella and varicella virus in serum samples were assessed by the LIAISON XL, a fully automated chemiluminescence analyzer, at seven different time points starting one week before HSCT and up to 12 months after HSCT. At baseline before HSCT most patients showed antibodies against measles (100%), mumps (80%), rubella (97,5%) and varicella (92,5%). 

Authors observed that although antibody titers declined over time, most patients retained antibodies against measles (92,5%), mumps (62,5%), rubella (87,5%) and varicella (85%) up to 12 months after HSCT. Acute and chronic Graft-versus-Host-Disease (GvHD) did not negatively influence persisting immunity against these diseases. 

Authors argue that considering that live attenuated vaccines should not be administered during the first year after HSCT, the persistence of antibodies against these diseases is of importance. Interestingly, acute and chronic GvHD did not significantly antibody titers.

Finally, authors conclude that antibodies against measles, mumps, rubella and varicella persist in most patients in the first year after HSCT. There was no significant difference between patients with acute or chronic GvHD and patients without GvHD concerning persistence of antibody titers.

Overall, the manuscript is highly valuable for the scientific community and should be accepted for publication.

======================

Other comments to authors:

1) Please check for typos throughout the manuscript.

2) With regards to Graft-versus-Host-Disease (GvHD) – authors are kindly encouraged to cite the following article that describes certain aspects of GvHD related to HSCT. DOI: 10.3390/cancers13040743

Author Response

1) Please check for typos throughout the manuscript.

We checked for typos in the manuscript.

2) With regards to Graft-versus-Host-Disease (GvHD) – authors are kindly encouraged to cite the following article that describes certain aspects of GvHD related to HSCT. DOI: 10.3390/cancers13040743

We cited the article in the introduction as requested.

Reviewer 4 Report

This study focuses on the levels of antibodies against measles, mumps, rubella and varicella in previously immunized patients during the first year after hematopoietic stem cell transplantation.  The study was conducted to evaluate the potential susceptibility of HSCT patients to these diseases, since the administration of such live-attenuated vaccines to these immunocompromised patients is advised against.  The authors present very strong data (with only a few minor exceptions) in support of their conclusion that the levels of these antibodies persist in the year following the onset pf HSCT.  Similarly, neither acute nor chronic graft-vs-host-disease significantly altered antibody levels.

This study is considered quite convincing and certainly reassuring to patients undergoing HSCT.  However, there are some criticisms of the presentation, primarily concerning relatively minor errors of omission, that should be addressed in a revised manuscript:

1)    It would be helpful for the reader to be given some indication of the criteria for grade 1-4 acute GVHD;

2)    What is the unit of time for the timepoints T-1 to T+5 ?

3)    The authors should comment as to why 10/21 male, but only 2/19 female, patients were autologous.  Couldn’t this skew the results with respect to gender?

Author Response

1)    It would be helpful for the reader to be given some indication of the criteria for grade 1-4 acute GVHD;

Grading of acute GvHD is based on the severity of skin involvement (extension of maculopapular rash and/or bullous lesions), the serum bilirubin levels and the severity of diarrhoea. We added a reference of a consensus paper about aGvHD grading.

Change in manuscript (line 98-100):

Grading of acute GvHD is based on the severity of skin involvement (extension of maculopapular rash and/or bullous lesions), the serum bilirubin levels and the severity of diarrhea as defined by Glucksberg.

Przepiorka, D.; Weisdorf, D.; Martin, P.; Klingemann, H.G.; Beatty, P.; Hows, J.; Thomas, E.D. 1994 Consensus Conference on Acute GVHD Grading. Bone marrow transplantation 1995, 15, 825-828.

2)    What is the unit of time for the timepoints T-1 to T+5?

T -1 to T +5 represent a certain time point which is described in the methods section (line 88-91). Serum samples were collected at the following time points: on admission (7 days be-fore HSCT; T-1), on the day of HSCT (T0), during aplasia (absolute neutrophil count <0.5 G/L; T+1), at engraftment (T+2, defined by a neutrophil count >0.5G/l), 1 month after HSCT (T+3), 3 to 6 months after HSCT (T+4), and 6 to 12 months after HSCT (T+5). We added this information to Figure 1.

3)    The authors should comment as to why 10/21 male, but only 2/19 female, patients were autologous.  Couldn’t this skew the results with respect to gender?

Of 12 autologous patients, 10 were male and only 2 were female. We agree with the reviewer that this could have influenced the results to a certain degree. The underlying diseases treated by autologous transplantation (lymphoma, myeloma) are more common in male than in female patients (about 3:2) which might have influenced the recruitment of patients. Female gender was a significant predictor for higher titres at T+5 for measles but not for the other three diseases. We acknowledge this limitation in the Discussion section.

Change in manuscript (line 306-08 and 318-21):

Considering that the autologous control group was predominantly male (10 of 12 patients), this might have influenced our results to a certain degree.

Possible weaknesses of the study are the small number of patients in each group, the predominantly male control group, the retrospective design and the lack of information on vaccination status.

Round 2

Reviewer 2 Report

The authors fully replied to the questions.

..5) The authors should check the dots in the figures along the text. ......This was raised because Figure 2 is unclear about the represented dots.